# PR3-ANCA Positive Behçet’s Disease with Severe Multisystem Involvement: A Diagnostic Challenge

**DOI:** 10.3390/diagnostics15222897

**Published:** 2025-11-15

**Authors:** Aleksandra Plavsic, Snezana Arandjelovic, Uros Karic, Jelena Ljubicic, Jovana Stanisavljevic, Adi Hadzibegovic, Dragan Vasin, Sergej Marjanovic, Rada Miskovic

**Affiliations:** 1Clinic for Allergy and Immunology, University Clinical center of Serbia, 11000 Belgrade, Serbia; snezanaaa2014@gmail.com (S.A.); jelena.ljubicic.mfub@gmail.com (J.L.); rada_delic@hotmail.com (R.M.); 2Faculty of Medicine, University of Belgrade, 11000 Belgrade, Serbia; uroskaric@gmail.com (U.K.); jovanastanisavljevic86@gmail.com (J.S.); a.hadzibegovic@gmail.com (A.H.); draganvasin@gmail.com (D.V.); sergejmarjanovic01@gmail.com (S.M.); 3Clinic for Infectious and Tropical Diseases, University Clinical center of Serbia, 11000 Belgrade, Serbia; 4Department of Anesthesiology, Emergency Center, University Clinical Center of Serbia, 11000 Belgrade, Serbia; 5Emergency Radiology Department, University Clinical Center of Serbia, 11000 Belgrade, Serbia

**Keywords:** Behçet’s disease (BD), granulomatosis with polyangiitis (GPA), myocardial infarction with non-obstructive coronary arteries (MINOCA), PR3-ANCA, intestinal perforation

## Abstract

**Background:** Behçet’s disease (BD) and granulomatosis with polyangiitis (GPA) are distinct vasculitides. PR3-ANCA is considered specific for GPA, yet rare BD cases demonstrate positivity, creating diagnostic dilemmas. **Case Presentation:** We describe a young man fulfilling criteria for BD, presenting with recurrent oral and genital ulcers, ocular inflammation, catastrophic jejunal perforations, pulmonary embolism, and myocardial infarction with non-obstructive coronary arteries. Despite strong PR3-ANCA positivity, the global phenotype was consistent with BD. Management required a complex, multimodal immunosuppressive regimen that included corticosteroids, cyclophosphamide, therapeutic plasma exchange, and rituximab. **Conclusions:** PR3-ANCA positivity may represent a severe BD phenotype rather than true GPA overlap, underscoring the need for individualized treatment strategies.

## 1. Introduction

Granulomatosis with polyangiitis (GPA) and Behçet’s disease (BD) are rare systemic vasculitides characterized by distinct epidemiological, clinical, and immunological profiles. GPA is an ANCA-associated vasculitis characterized by necrotizing granulomatous inflammation of small to medium vessels, often affecting the respiratory tract and kidneys [1]. BD, in contrast, is a variable-vessel vasculitis defined by recurrent oral and genital ulcers, ocular inflammation, skin lesions, gastrointestinal ulceration, and vascular complications [2]. BD shows a strong association with HLA-B*B51, which confers increased genetic susceptibility and is associated with more severe manifestations. Although PR3-ANCA positivity is considered highly specific for GPA, rare reports have described its occurrence in BD, creating diagnostic dilemmas. The reported frequency of ANCA positivity in BD varies considerably across cohorts (ranging from 2.2% to 10.2%) [3,4]. Both GPA and BD share pathogenic features, including neutrophil hyperactivation, endothelial dysfunction, and complement activation. Classification criteria for systemic vasculitides continue to evolve, with the 2022 ACR/EULAR criteria and updated International Criteria for Behçet’s Disease (ICBD) criteria improving diagnostic sensitivity [5,6]. Here, we present a patient with life-threatening BD fulfilling ICBD criteria and strong PR3-ANCA positivity, highlighting diagnostic ambiguity, therapeutic challenges, and the need for flexible classification frameworks.

## 2. Case Report

A 30-year-old man with no previous medical history presented with rhinorrhea, nasal congestion, frontal headache, fever, general malaise, polyarthralgia, and non-progressive cough lasting for approximately 10 days. He was initially treated with oral amoxicillin, but symptoms worsened, and he was admitted to the Infectious Diseases hospital. On admission, he had fever (39 °C), tachycardia, recurrent oral and genital ulcers, left eye iridocyclitis, and nasal septum perforation with abundant serohemorrhagic discharge. CT of the paranasal sinuses showed mucosal thickening; chest CT revealed right lower lobe consolidation. Laboratory markers were markedly elevated (CRP > 220 mg/L, fibrinogen 7 g/L, WBC 39 × 10^9^/L, neutrophils 25 × 10^9^/L, platelets 753 × 10^9^/L). Extensive microbiology testing was negative. Urinalysis revealed erythrocyturia and proteinuria (0.5 g/24 h). Immunology showed strong c-ANCA positivity (>1:640) and PR3-ANCA > 200 IU/mL. HLA typing showed HLA-B*51 positivity. The constellation of oral and genital ulcers, ocular inflammation, HLA-B*51 positivity, and ANCA positivity raised suspicion for overlap of BD with GPA. High-dose methylprednisolone pulses (500 mg for 3 days) were administered, followed by 2 mg/kg/day, resulting in a reduction in systemic symptoms and partial ulcer resolution. Shortly thereafter, the patient developed chest tightness with ST elevation and increase in troponinlevel. Coronary angiography revealed no obstructive lesions, and myocardial infarction with non-obstructive coronary arteries (MINOCA) was diagnosed, most likely attributed to coronary vasculitis. Dual antiplatelet therapy and low-molecular-weight heparin (LMWH) were added. Cyclophosphamide pulse (400 mg) was initiated (an additional fractionated 600 mg pulse was planned), but on the same day, severe abdominal pain developed. We performed abdomen CT with angiography after intravenous contrast administration in the arterial and venous phases (“bolus tracking”), with multiplanar reconstructions (MPR) at 1 mm slice thickness. CT angiography showed pneumoperitoneum, jejunal wall edema, and ischemia (Figure 1).

Emergency laparotomy revealed four jejunal perforations with less than 200 cm of viable small bowel remaining; jejunostomy was performed. Histopathology was unfortunately not obtained. Postoperatively, seven therapeutic plasma exchange (TPE) sessions were given with methylprednisolone (2 × 500 mg/day), cyclophosphamide (600 mg), and broad-spectrum antibiotics. Short bowel syndrome developed, with 25 kg weight loss. Endoscopic jejunal biopsy performed seven days after the surgical procedure showed lymphoplasmacytic infiltrates, focal neutrophils, erosive lesions, and villous flattening without granulomas on hematoxylin/eosin staining protocol.

Rituximab (375 mg/m^2^ weekly × 4) was administered, with marked clinical and laboratory improvement and decline of PR3-ANCA antibody titers. However, several months later, the patient developed acute pulmonary embolism; CT angiography confirmed thrombi in right interlobar/lobar and left segmental arteries, without aneurysms or stenoses. We also performed cardiac MRI in standard imaging planes using functional TrueFISP sequences and morphological TSE sequences in three planes, both without contrast administration. Following contrast administration, imaging was performed using IR sequences and T1 and T2 mapping with the MOLLI sequence. Cardiac MRI showed reduced EF (45%), regional wall motion abnormalities, and subendocardial fibrosis (7% LV mass) with a decreased extent compared to the previous MRI finding. Cyclophosphamide pulses were reinitiated at a dose of 800 mg, reaching a cumulative dose of 5.8 g to date. Planned bowel reconstruction remains delayed. During a six-month hospitalization, the patient faced significant challenges in achieving calorie and protein goals, resulting in a weight loss of 25 kg from a baseline weight of 105 kg. Early postoperative nutrition was primarily provided through parenteral nutrition, followed by the initiation of trophic enteral nutrition on the third postoperative day, which was gradually increased alongside vitamin and oligoelement supplementation, as well as the use of loperamide. After two weeks, nutrition was provided orally with oral nutritional supplements and food fortification, with nutritional goals set at 30 kcal/kg/day and 1.5 g/kg/day of protein. The patient was a candidate for bowel reconstruction and not for home parenteral nutrition, did not lose more weight, and experienced clinical improvement. Although overlap with GPA was initially suspected due to strong PR3-ANCA positivity and ENT involvement, the subsequent evolution with recurrent mucocutaneous ulcers, ocular inflammation, catastrophic gastrointestinal perforations, thromboembolic events, and cardiac involvement was more consistent with severe BD, leading to the interpretation of this case as PR3-ANCA positive BD rather than a true overlap. The patient is currently considered to be in pharmacological remission; however, long-term and regular follow-up remains necessary. Key events are summarized in the clinical timeline (Figure 2).

## 3. Discussion

This case illustrates the diagnostic and therapeutic complexity of systemic vasculitis when classical serological markers do not align with the dominant clinical phenotype. Although the patient exhibited strong PR3-ANCA positivity, a feature considered highly specific for GPA, the constellation of recurrent oral and genital ulcers, ocular inflammation, severe gastrointestinal perforations, pulmonary embolism, and MINOCA strongly supported BD as the underlying diagnosis.

The coexistence or overlap of GPA and BD is exceptionally rare, with only sporadic case reports described in the literature [7,8,9]. Although ANCA are considered highly specific for ANCA-associated vasculitides, several studies have reported their presence in BD. In a study of 808 patients with BD diagnosed according to the 2014 ICBD criteria, Kim et al. reported ANCA positivity in 2.2% of patients. This was associated with vascular involvement of the upper extremities and visceral arteries; however, it did not predict adverse outcomes during follow-up, including deep vein thrombosis, acute coronary syndrome, or all-cause mortality [3]. Case reports have further highlighted overlap syndromes between BD and GPA, where patients demonstrated PR3-ANCA positivity, raising important diagnostic and therapeutic challenges [10,11,12,13,14]. These findings suggest that while ANCA positivity in BD is uncommon, its occurrence, particularly PR3-ANCA, may reflect either an overlap with GPA or atypical immunological activation within BD.

At the immunological level, GPA and BD share several converging pathways. In GPA, PR3-ANCA antibodies bind to neutrophil surface antigens, inducing degranulation, the release of reactive oxygen species, and the formation of neutrophil extracellular traps (NETs) that contribute to endothelial injury and necrotizing vasculitis [15,16]. Similarly, BD is characterized by spontaneous neutrophil activation and excessive NETosis, driven by pro-inflammatory cytokines such as IL-1β, IL-6, and TNF-α, which perpetuates vascular inflammation and thrombosis [17]. This shared neutrophil-driven pathology may explain the overlapping clinical features seen in our patient, such as mucocutaneous ulcerations, vasculitic bowel ischemic injury, and thromboembolic events. Endothelial dysfunction further compounds vascular injury in both conditions. Activated neutrophils induce endothelial cell damage directly and through the upregulation of adhesion molecules (VCAM-1, ICAM-1), facilitating leukocyte recruitment and inflammation [18,19]. Complement activation, particularly the alternative pathway, has been implicated in GPA pathogenesis and is increasingly recognized in BD, to a lesser extent [19]. Complex immunopathology may manifest clinically as variable-vessel vasculitis, spanning small to large vessels, as demonstrated in our patient’s diverse organ involvement. Taken together, these mechanisms provide a biological rationale for why BD patients may occasionally present with ANCA positivity, not as true overlap but as an epiphenomenon of neutrophil-driven inflammation.

Clinically, the dominant features in our case favored BD. Gastrointestinal perforations are exceedingly rare in GPA (<10%) but are a well-recognized, sometimes life-threatening complication of BD, particularly in young men [20,21]. The presence of multiple jejunal perforations in our patients suggests severe small-vessel vasculitis, likely reflecting additive or synergistic effects of overlapping inflammatory cascades. Histopathological confirmation was unfortunately not obtained. The histopathological diagnosis of intestinal vasculitis remains challenging. The submucosal or transmural vessel involvement typical of GPA and BD is often missed on superficial biopsies. This highlights a major diagnostic gap: standard gastrointestinal biopsies often miss pathognomonic features due to sampling limitations, especially in vasculitides localized to submucosal vessels. Rapid progression to intestinal ischemia and multiple perforations in our patient within a short therapeutic window underscored the fulminant nature of the inflammatory vasculitis, warranting immediate surgical and immunosuppressive intervention. We decided to perform TPE and intense immunosuppressive management, which was critical to achieving stable conditions and preventing further complications.

Cardiac involvement, including coronary arteritis and MINOCA, is also more typical of BD than GPA [22,23]. Myocardial infarction with MINOCA represents a diagnostic and therapeutic challenge, as illustrated in our patient. The absence of significant coronary stenosis on angiography, combined with cardiac MRI findings of ischemic fibrosis, strongly supports vasculitic coronary involvement, possibly mediated by microvascular inflammation and thrombosis. Previous reports have documented similar phenomena in ANCA-associated vasculitis, highlighting the need for heightened clinical suspicion [1,22]. Also, the use of cardiotoxic drugs, such as cyclophosphamide, had to be carefully balanced. This was the reason for fractionated cyclophosphamide dosing in the initial treatment of our patient.

A significant, additional manifestation in our patient was the pulmonary embolism (PE), which developed after intense immunosuppressive and biologic therapy. Thromboembolic events are well-recognized in BD and less frequently in GPA, often reflecting endothelial injury, hypercoagulability, and inflammation-driven thrombogenesis [24]. Management of thrombosis in the context of active vasculitis is complex, balancing anticoagulation to prevent further embolism against the risk of bleeding. In our patient, continuation of cyclophosphamide and anticoagulation was critical, guided by a multidisciplinary team.

Therapeutically, our case demonstrates the utility of multimodal immunosuppression combining high-dose corticosteroids, cyclophosphamide, TPE, and rituximab, with prompt decision making. High-dose corticosteroids and cyclophosphamide were given as first-line therapy, reflecting concern for fulminant ANCA-associated vasculitis. Plasma exchange was added as salvage therapy after severe intestinal perforations, consistent with guidelines for life-threatening vasculitis flares. Rituximab, a B-cell-depleting agent widely used in refractory GPA, was subsequently introduced. Although not licensed for BD, several reports, including systematic reviews, have documented benefit in severe and refractory cases, particularly those with ocular or vascular involvement [25,26]. In our patient, rituximab was associated with substantial clinical improvement and reduction in PR3-ANCA antibody titers. This aligns with the hypothesis that B cells may contribute to BD pathogenesis, either directly or via crosstalk with neutrophils and T cells. Management of short bowel syndrome and nutritional rehabilitation added complexity, necessitating a multidisciplinary approach involving surgery, nutrition, immunology, and rehabilitation teams. The patient’s gradual clinical and immunologic improvement highlights the potential for recovery even in severe vasculitis syndromes when managed promptly and comprehensively. Beyond rituximab, other biologics are emerging in BD management. TNF-α inhibitors remain the cornerstone for refractory cases. IL-1 inhibitors (anakinra, canakinumab), IL-6 blockade (tocilizumab), and JAK inhibitors have shown promise in recent clinical studies, particularly in patients with gastrointestinal and vascular involvement [27,28,29]. Our case highlights that in life-threatening presentations, clinicians may need to combine conventional cytotoxic drugs with biologics, tailoring therapy to the individual patient’s phenotype rather than rigid diagnostic categories. These discrepancies highlight the limitations of current classification criteria. The ICBD criteria have high sensitivity and specificity, and in this case yielded a score of 7 points (recurrent oral and genital ulcers (4 pts.), iridocyclitis (2 pts.), vascular manifestations (1 pt.)), well above the diagnostic threshold [5]. In contrast, reliance on ANCA positivity, as emphasized in the 2022 ACR/EULAR GPA criteria, risks misclassifying such patients (6 diagnostic features of the patient compared to BD, GPA, and overlap are presented in Table 1).

A recent 2025 multicenter study demonstrated that nearly 9% of 280 BD patients initially labeled as overlap syndromes (OS-BD-AAV) were reclassified after longitudinal follow-up, underscoring the fluidity of vasculitis phenotypes [30]. Most of these patients were MPO-ANCA positive with microscopic polyangiitis features, while others were PR3-ANCA positive with GPA features. These findings support the notion that systemic vasculitides should be viewed as a spectrum, where the clinical expression may shift over time, and serology does not always correspond to the phenotype. As we previously mentioned, the frequency of ANCA positivity in BD varied across articles (2.2–10.2%). A possible explanation for this discrepancy lies in the methodological differences in antibody detection. In studies reporting lower ANCA positivity rates, indirect immunofluorescence (IIF) was used as the initial screening method, with subsequent ELISA testing for specific antigens (MPO and PR-3) performed only in IIF-positive patients [3]. In contrast, a study reporting higher prevalence applied both IIF and ELISA for all samples and demonstrated that certain ANCA, particularly antibodies against bactericidal/permeability-increasing protein (BPI), may be negative by IIF [4].

Therefore, comprehensive ELISA testing for specific antigens or extended ANCA profiling (including BPI, proteinase 3, myeloperoxidase, elastase, cathepsin G, and lactoferrin) is crucial in such cases. Through this case report, we aimed to emphasize the clinical relevance of ANCA antibody testing in BD patients, as their positivity may serve as a useful indicator warranting increased vigilance for the development of vascular and gastrointestinal complications.

Another strength of this report lies in the detailed documentation of a rare and severe presentation of BD with misleading ANCA serology. A few reports described gastrointestinal perforations, MINOCA, and PE in one patient. The case also emphasizes the importance of multidisciplinary collaboration of immunologists, rheumatologists, infectious disease specialists, anesthesiologists, and surgeons. Limitations include the absence of histopathological confirmation from the resected bowel, which might have clarified the presence or absence of granulomatous inflammation. A renal biopsy was not performed, leaving open the possibility of subclinical GPA-related nephritis. Furthermore, the current follow-up period is one year, and the long-term efficacy of rituximab in this phenotype is unknown. Thus, while ANCA serology raised suspicion for GPA/BD overlap, the overall phenotype remains consistent with severe vascular BD.

## 4. Conclusions

This case supports the growing recognition that PR3-ANCA positivity does not exclude BD and may instead signal a severe vascular phenotype within BD. Systemic vasculitides should be conceptualized as a continuum, where immunopathogenesis and clinical features guide diagnosis and therapy. Individualized treatment, including biologics traditionally reserved for ANCA-associated vasculitis, may be warranted in severe BD presentations. Clinicians should remain cautious when interpreting ANCA serology and prioritize the clinical phenotype when making diagnostic and therapeutic decisions.

## Figures and Tables

**Figure 1 diagnostics-15-02897-f001:**
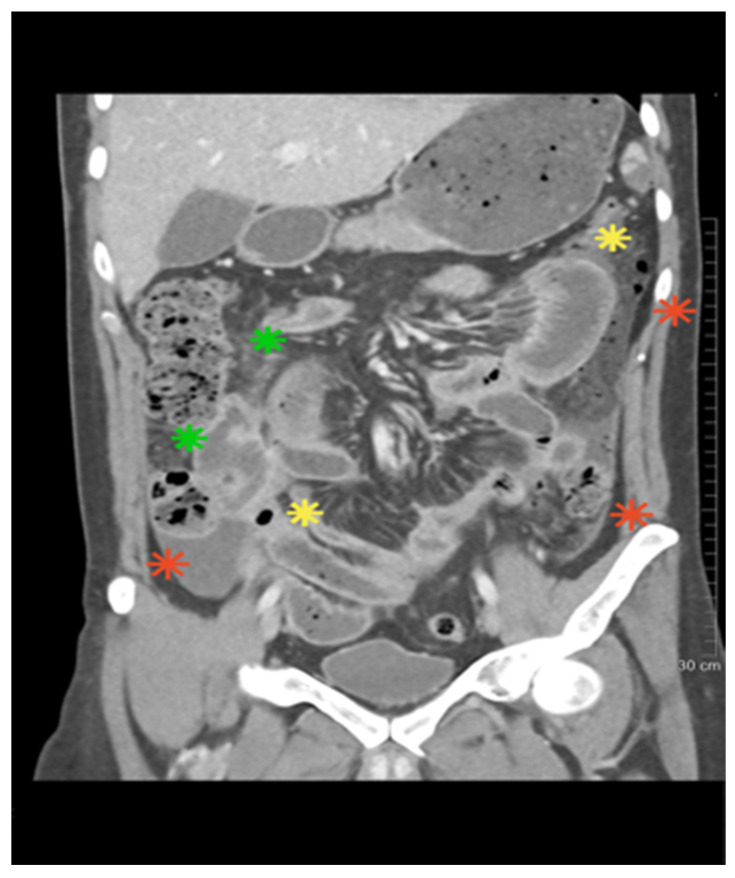
The abdominal CT scan revealed the presence of bowel loop ischemia (green), pneumoperitoneum (yellow), and fluid accumulation in the peritoneal recesses (red).

**Figure 2 diagnostics-15-02897-f002:**
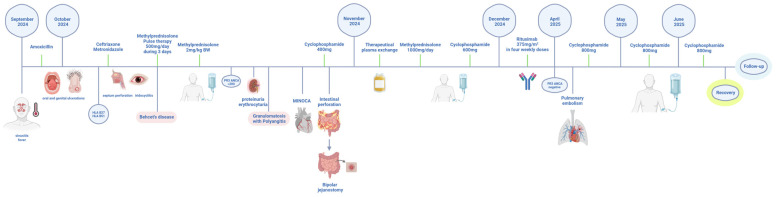
Timeline of the case report. Created in BioRender. Ljubicic, J. (2025) https://BioRender.com/x1070jj (accessed on 12 November 2025) and licensed under CC BY 4.0.

**Table 1 diagnostics-15-02897-t001:** Diagnostic features of the patient compared to BD, GPA, and overlap.

Manifestations/Criteria	Behçet’s Disease (ICBD 2014)	Granulomatosis with Polyangiitis (ACR/EULAR 2022)	Patient (Case Report)	Interpretation-Dominant Phenotype
Oral ulcers	Mandatory (recurrent, +2 points)	Not typical	Recurrent	Strongly supports BD
Genital ulcers	High specificity (+2)	Rare	Present	Strongly supports BD
Ocular involvement	Typical (uveitis, retinal vasculitis)	Possible but less common	Iridocyclitis	Supports BD
Skin lesions	Erythema nodosum, pustules (+1)	Nonspecific	Absent	Neutral
Vascular manifestations	Venous/arterial thrombosis, aneurysms	Necrotizing vasculitis also possible	Pulmonary embolism, MINOCA	More typical of BD
Gastrointestinal perforations	Characteristic, esp. ileocecal/jejunal	Exceedingly rare (<10%)	4 jejunal perforations	Decisive for BD
ENT involvement	Rare	Very typical (sinusitis, granulomas, septal perforation)	Sinusitis, septal perforation	Supports GPA
Pulmonary lesions	Rare	Common (nodules, alveolar hemorrhage)	Absent	Against GPA
Renal involvement	Rare, mild	Frequent (GN, hematuria, proteinuria)	Microscopic hematuria, mild proteinuria	Mild GPA-like feature
HLA-B*51 positivity	Strong association	Not relevant	Positive	Supports BD
PR3-ANCA positivity	Rare (~2% of BD)	Hallmark of GPA	Strongly positive	Misleading, but not decisive
Overall phenotype	Fits BD criteria (≥4 points)	Partial GPA-like features	—	Severe BD with PR3-ANCA

Abbreviations: BD—Behçet’s Disease, GPA—Granulomatosis with Polyangiitis, ICBD—International Criteria for Behçet’s Disease, MINOCA—myocardial infarction with non-obstructive coronary arteries, GN—glomerulonephritis.

## Data Availability

The original contributions presented in this study are included in the article. Further inquiries can be directed to the corresponding author.

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
