# Peer review of "PR3-ANCA Positive Behçet’s Disease with Severe Multisystem Involvement: A Diagnostic Challenge"

_diagnostics, 2025, doi:10.3390/diagnostics15222897_

Round 1
Reviewer 1 Report
Comments and Suggestions for Authors
This case report may be potentially interesting enough for the journal's readers.
However, several concerns have arisen which should be addressed adequately.
- It is unclear the pathogenesis of PR3-ANCA positivity in this patient. Some scientific evidences regarding this issue should be represented. Unfortunately, coincidentally posistivity of PR3-ANCA in this patient cannot be excluded.
- Discussion section should be rewritten conicisely and logically focusing on the novel findings of this patient.
- Some tables which summarized this case and previously reported similar cases are necessary. Then, clear discussion similarities and differences between this case and previously reported similar cases.
- Novel aspect based on this case should be clarified.
Mmay be improved much concisely and logically.
Author Response
Thank you very much for taking the time to review this manuscript, for recognizing the content of our article, and for allowing us to make changes according to your suggestions. Please find the detailed responses below, and the corrections that were made are highlighted in yellow in the resubmitted file.
Thank you very much for taking the time to review this manuscript, for recognizing the content of our article, and for allowing us to make changes according to your suggestions. Please find the detailed responses below, and the corrections that were made are highlighted in yellow in the resubmitted file.
Response to the Reviewer 1:
- Comment 1: It is unclear the pathogenesis of PR3-ANCA positivity in this patient. Some scientific evidences regarding this issue should be represented. Unfortunately, coincidentally posistivity of PR3-ANCA in this patient cannot be excluded.
- Answer: In our study, we aimed to provide a detailed explanation of the crucial role of neutrophils in the pathogenesis of BD, based on all currently available data, while also offering potential mechanisms by which ANCA antibodies may contribute to the development of this condition (114–132). Although, without in vitro assays and functional studies, it is not possible to definitively determine whether ANCA antibodies exert a pathogenic effect in BD or only represent an accompanying phenomenon, previous studies (references 8, 10, 28) indicate that their presence cannot be ignored, particularly given the important phenotypic differences observed between ANCA-positive and ANCA-negative BD patients (further clarrifed in Answer to the Comment 3 of the second Reviewer).
- Comment 2: Discussion section should be rewritten conicisely and logically focusing on the novel findings of this patient.
- Answer: Thank you for this suggesrtion. We have made an effort to further enrich the Discussion section based on your suggestions as well as those of the second reviewer. The added parts are highlighted in yellow within the Discussion section.
- Comment 3: Some tables which summarized this case and previously reported similar cases are necessary. Then, clear discussion similarities and differences between this case and previously reported similar cases.
- Answer: Reported individual cases describing ANCA antibody positivity in BD are rare and often lack sufficient clinical and therapeutic details (references 10-14). Moreover, our study was not designed as a case-based review; therefore, we did not include a summary table of previously published cases, as recommended by the CABARET guidelines for case-based reviews. Instead, the structure of our report follows the CARE guidelines for case reports, which include a focused review of the literature in the discussion section to provide context.
- We consider of great relevance the study including a total of 84 participants, which we discussed in detail in our manuscript (121–127), where the characteristics of ANCA-positive BD patients were compared with our case. Notably, vascular manifestations were significantly more prevalent among ANCA-positive BD patients (22.2% vs. 6.1%), mirroring the dominant clinical presentation in our patient. The main complication observed during follow-up was the development of gastrointestinal involvement (12.4%), which was likewise seen in our patient.
- Comment 4: Novel aspect based on this case should be clarified.
- Answer: The major novelty of this case is the very rare occurrence of PR3-ANCA positivity in BD which may create significant diagnostic uncertainty.The BD presented with rare clinical features:catastrophic jejunal perforations, pulmonary embolism, and myocardial infarction with non-obstructive coronary arteries. Also, our report contributes to the limited literature by illustrating that PR3-ANCA positivity may reflect a severe phenotype of BD. Furthermore, the case highlights the complex, multidisciplinary and multimodal therapeutic approach, including the use of plasma exchange and rituximab. We have emphasized these aspects throughout the manuscript, particularly in the discussion section.
Reviewer 2 Report
Comments and Suggestions for Authors
Dear Authors
Thank you for submitting this important case report on a rare and clinically significant presentation. While the manuscript presents a valuable case, revisions are necessary to enhance its scientific clarity. Here I summarized my recommendations, strengths and limitations:
Strengths
Your case documents important clinical features: catastrophic jejunal perforations, MINOCA, pulmonary embolism, and PR3-ANCA positivity in severe BD. The clinical timeline (Figure 2) and comparative analysis (Table 1) are excellent, and the successful use of rituximab with ANCA titer decline provides valuable therapeutic evidence.
Revisions Required
- Introduction:
Please expand the literature review on ANCA prevalence in BD and include a more detailed mechanistic discussion on the potential reasons for ANCA positivity in BD patients.
- Methods - Critical Missing Details
- Provide step-by-step diagnostic criteria scoring (ICBD: show calculation for 8 points).
- Detail imaging protocols (CT specifications, cardiac MRI field strength, exact measurements)
- Clarify histopathology: When was endoscopic biopsy performed? What staining techniques were used?
- Specify exact follow-up duration (currently only stated as "limited")
- Case Presentation
- Discuss nasal septum perforation for GPA diagnosis
- Explain why renal biopsy was deferred despite hematuria/proteinuria
- Clarify the cyclophosphamide dosing regimen and cumulative dose
- Provide baseline weight and nutritional management details for a 25 kg weight loss - Discussion
- Address the histopathology gap more substantially—why was the surgical specimen tissue not obtained?
- Expand rationale for rituximab selection over other biologics
- Provide specific follow-up outcomes: current ANCA status, remission status, ongoing immunosuppression
- Discuss why ENT/renal findings didn't favour GPA diagnosis
Thanks again for reporting this case, and looking forward to reviewing the revised manuscript.
With Kind Regards
Author Response
Thank you very much for taking the time to review this manuscript, for recognizing the content of our article, and for allowing us to make changes according to your suggestions. Please find the detailed responses below, and the corrections that were made are highlighted in yellow in the resubmitted file.
- Comment 1: Introduction: Please expand the literature review on ANCA prevalence in BD and include a more detailed mechanistic discussion on the potential reasons for ANCA positivity in BD patients.
- Answer: Thank you for this comment.We have added the prevalence of ANCA positivity among patients with BD in the Introduction, and in the Discussion we provided a detailed explanation of the variability in ANCA positivity depending on the detection method used (226-237). We have also mentioned the potential mechanism of ANCA antibody action in BD in the Introduction, while a detailed explanation is provided in the Discussion section (lines 135-153).
- Comment 2: Provide step-by-step diagnostic criteria scoring (ICBD: show calculation for 8 points).
- Answer: We apologize for this inconvinience. The precise ICBD score for our patient is 7 and we have added calculation in our article: reccurent oral and genital ulcers (4 pts.), iridocyclitis (2 pts.), vacular manifestations (1 pt.) (lines 210-213)
- Comment 3: Detail imaging protocols (CT specifications, cardiac MRI field strength, exact measurements).
- Answer: Thank you for this suggestion. We have added abovementioned specifications about CT scan (lines 70-72) and cardiac MRI (lines 89-92).
- Comment 4 and 5: Address the histopathology gap more substantially—why was the surgical specimen tissue not obtained? Clarify histopathology: When was endoscopic biopsy performed? What staining techniques were used?
- Answer: Given that the patient was in a life-threatening condition at that time, an emergency laparotomy with the creation of a jejunostomy was performed. The surgery was carried out overnight in the emergency department; therefore, it was not possible to request a tissue sample for histopathological analysis at the time of the intervention, and unfortunately, no sample was obtained by the surgical team. An endoscopic biopsy was performed seven days after the surgical intervention, once the patient’s overall condition had stabilized. Staining techinque used for analyisis was H&E (hematoxylin and eosin). (lines 82-84)
- Comment 6: Specify exact follow-up duration (currently only stated as "limited")
- Answer: In lines 248-249 we have added duration of follow-up period of one year at this moment.
- Comment 7: Discuss nasal septum perforation for GPA diagnosis.
- Answer: Various manifestations of nasal mucosal involvement in BD have previously been described, including nasal obstruction, crusting, discharge, and ulceration (Shahram F, Zarandy MM, Ibrahim A, et al. Nasal Mucosal Involvement in Behcet Disease: A Study of its Incidence and Characteristics in 400 Patients. Ear, Nose & Throat Journal. 2010;89(1):30-33. doi:10.1177/014556131008900109), with septal perforation occurring only in rare cases (Meyer A, Czerny M, Antisdel J. Nasal mucosa ulceration and septal perforation as initial presentation of a patient with probable Behcet’s disease. Int J Pediatr Otorhinolaryngol Extra 9: 56-59, 2014). Additionally, although not mentioned in the manuscript, a biopsy of the nasal septal mucosa was performed. In the examined sample, there was infiltration by polymorphonuclear cells with fibrin deposits, without features typical of GPA, such as granulomatous inflammation or extensive necrosis. When considered in the context of the patient’s overall clinical presentation, the nasal septum perforation in our patient was more consistent with BD than with GPA.
- Comment 8: Explain why renal biopsy was deferred despite hematuria/proteinuria.
- Answer: Although our patient initially presented with proteinuria and hematuria, these clinical findings did not recur during follow-up, and therefore the nephrologist did not consider a renal biopsy indicated.
- Comment 9: Clarify the cyclophosphamide dosing regimen and cumulative dose.
- Answer: Cyclophosphamide was administered in fractionated doses, initially due to the patient’s MINOCA findings, and later in response to the cardiac MRI results, in the context of the well-known cardiotoxic effects on the myocardium, as discussed in our manuscript. (lines 176-178) Cumulative dose was added in approprate part of the article. (lines 94-95)
- Comment 10: Provide baseline weight and nutritional management details for a 25 kg weight loss.
- Answer: Thank you for this valuable comment. We have added explanation of nutritional management which was carried out by a specialized team of anesthesiologists and nutritionists in the postoperative period. (lines 96-105).
- Comment 11: Expand rationale for rituximab selection over other biologics
- Answer: Although TNF inhibitors are considered first-line therapy for BD patients unresponsive to conventional treatments (e.g, cyclophosphamide), we chose to treat our patient with rituximab since initially he was considered as a potential overlap between BD and GPA, and ultimately our decision was made by the markedly elevated PR3-ANCA levels, indicating a pathogenic mechanism predominantly driven by B-cell activity.
- Comment 12: Provide specific follow-up outcomes: current ANCA status, remission status, ongoing immunosuppression.
- Answer: At this time, after one year of follow-up, ANCA is negative by IIF, and PR3-ANCA is <2 IU/mL. The patient is receiving a daily dose of methylprednisolone at 0.4 mg/kg along with monthly cyclophosphamide pulses (800 mg). The patient is currently considered to be pharmacological remission; however, long-term and regular follow-up remains necessary.
- Comment 13: Discuss why ENT/renal findings didn't favour GPA diagnosis.
Answer: We agree that the nasal septum perforation, along with the initial presence of renal involvement, and PR3-ANCA positivity, suggested a possible diagnosis of GPA in our patient. However, the patient did not exhibit other typical features of GPA, such as pulmonary involvement, upper respiratory tract disease, or histopathological confirmation of granulomas, and no renal involvement was observed during follow-up. The most important events in his clinical presentation were oral and genital ulcers, vascular manifestations, intestinal perforation, alongside HLA-B51 positivity. Therefore, we consider that the overall clinical presentation is more consistent with BD than with GPA
Round 2
Reviewer 1 Report
Comments and Suggestions for Authors
I understnad the authors' responses. The revised MS has been addressed well.
Comments on the Quality of English LanguageMmay be improved much concisely and logically.